# A prospective evaluation of Johnson & Johnson COVID-19 vaccine on glycemic biomarkers in type 2 diabetes mellitus in Ethiopia

Chala Kenenisa Edae[1]*, Abdisa Tufa Bedada[1], Maria Degef Teklemariam[1], Mesfin Bekele Tolesa[2], Solomon Genet Gebre[1]

**1** Department of Biochemistry, School of Biomedicine and Laboratory Science, Addis Ababa University, Addis Ababa, Ethiopia, **2** Director at Adama Public Health Referral Laboratory and Research Center, Adama, Ethiopia

* chala.kenenisa@ju.edu.et

## Abstract

### Background

People living with Type 2 Diabetes Mellitus (T2DM) face a heightened risk of experiencing severe complications from COVID-19, underscoring the importance of vaccination. Nonetheless, the impact of COVID-19 vaccines—especially the Johnson & Johnson (Ad26.COV2.S) vaccine—on glucose regulation has not been fully elucidated. This study evaluates the impact of vaccination on glycemic parameters, including Random Blood Sugar (RBS) and Hemoglobin A1c (HbA1c), and identifies factors influencing glycemic variability.

### Methods

Between May 2023 and June 2024, a prospective cohort study was carried out at Adama Hospital Medical College in Ethiopia. Adults diagnosed with Type 2 Diabetes Mellitus were divided into two cohorts based on vaccination status: those who received the vaccine and those who did not. Glycemic parameters were recorded at baseline and subsequently at three-month intervals—specifically at 3, 6, 9, and 12 months following vaccination. To evaluate trends over time and identify influencing factors, including demographic and clinical variables, longitudinal data were analyzed using Generalized Estimating Equations (GEE).

### Results

Vaccinated individuals exhibited transient elevations in RBS, peaking at three months post-vaccination before stabilizing. In contrast, HbA1c levels demonstrated a gradual increase over time. Greater glycemic variability was observed in younger individuals and females. The primary determinants of variations in glycemic levels were

**Data availability statement:** All relevant data are within the paper and its Supporting Information files.

**Funding:** The author(s) received no specific funding for this work.

**Competing interests:** The authors have declared that no competing interests exist.

vaccination status, duration following immunization, and demographic characteristics. In contrast, diabetes treatments and lifestyle-related factors showed only a limited influence.

## Conclusion

The Johnson & Johnson COVID-19 vaccine was associated with short-term RBS fluctuations and a sustained increase in HbA1c levels in T2DM patients. These findings highlight the need for personalized glycemic monitoring post-vaccination. Despite these metabolic variations, the vaccine's protective role against severe COVID-19 outweighs transient glycemic disturbances. Incorporating vaccination efforts into comprehensive diabetes management is crucial, and additional studies are warranted to investigate the underlying biological mechanisms.

## 1. Introduction

Diabetes mellitus (DM) is a chronic condition characterized by elevated blood glucose levels resulting from insufficient insulin production or ineffective insulin action [1]. The worldwide burden of Type 2 Diabetes Mellitus (T2DM) is steadily increasing, with estimates suggesting that the number of affected individuals could reach 643 million by the year 2030 [2]. Ethiopia, like other low- and middle-income countries, faces an increasing burden of diabetes due to urbanization, lifestyle changes, and genetic predisposition [3].

In 2021, the estimated prevalence of diabetes in Ethiopia was 3.2%; however, due to widespread underdiagnosis and restricted access to medical services, the actual rate is likely higher [4]. Type 2 Diabetes Mellitus is linked to several chronic complications, including cardiovascular disorders, nerve damage, kidney impairment, and vision-related issues such as retinopathy [5]. Additionally, it increases susceptibility to infections, including severe COVID-19, which has further strained healthcare systems worldwide [6,7].

COVID-19 vaccines, including the Johnson & Johnson (Ad26.COV2.S) vaccine, have been instrumental in reducing severe disease and mortality [8,9]. However, emerging evidence suggests that vaccines may impact metabolic parameters, particularly glycemic control, in individuals with T2DM [10,11]. The immune reaction initiated by vaccination results in the secretion of pro-inflammatory cytokines, including interleukin-6 (IL-6) and tumor necrosis factor-alpha (TNF-α). These cytokines have the potential to disrupt insulin signaling pathways, thereby causing a temporary elevation in blood glucose levels [12]. Furthermore, the stimulation of the hypothalamic-pituitary-adrenal (HPA) axis following vaccination can cause a rise in cortisol secretion, which in turn enhances gluconeogenesis and reduces the body's sensitivity to insulin [13]. These mechanisms contribute to temporary disruptions in glycemic control, particularly in individuals with preexisting diabetes.

Recent studies confirm that glycemic control influences vaccine immunogenicity. Marfella et al. (Nat Commun, 2022) [14] demonstrated that poor glycemic control is

linked with higher SARS-CoV-2 breakthrough infections among patients with type 2 diabetes mellitus (T2DM). Sardu et al. (J Clin Med, 2022) [15] explained that hyperglycemia impairs vaccine effectiveness. The CAVEAT study (Marfella et al., Diabetes Obes Metab, 2022) [16] confirmed reduced immunogenicity in patients with poor glycemic control [17,14,15].

Moreover, studies suggest that chronic hyperglycemia may impair vaccine efficacy by reducing antibody production and weakening T-cell activation, thereby affecting immune responses [12]. Given that effective glycemic control is crucial for reducing both acute and long-term complications in diabetes [18], understanding the metabolic effects of COVID-19 vaccination in diabetic populations is essential. Despite widespread vaccination campaigns, limited data exist on how the Johnson & Johnson vaccine affects glycemic stability in individuals with diabetes, particularly in resource-limited settings such as Ethiopia. Although previous research has highlighted vaccine-induced metabolic changes, the specific effects on glycemic control in African populations, including Ethiopia, remain largely unexplored. The interplay between vaccine-induced inflammation, insulin resistance, and long-term glycemic trends in diabetic individuals has not been adequately studied in real-world clinical settings. Addressing this knowledge gap is essential for optimizing diabetes management strategies in vaccinated populations.

This prospective cohort study is designed to evaluate the impact of the Johnson & Johnson COVID-19 vaccine on glycemic regulation in individuals with type 2 diabetes mellitus (T2DM) by monitoring alterations in Random Blood Sugar (RBS) levels and Hemoglobin A1c (HbA1c) values over a period of time. By comparing vaccinated and unvaccinated individuals, this study will provide evidence-based insights to inform diabetes management in Ethiopia and similar healthcare settings.

## 2. Materials and methods

### 2.1. Study design and setting

This study employed a prospective observational design to evaluate the immunological response following COVID-19 vaccination with the Johnson & Johnson vaccine among adult participants in Adama Hospital Medical College in Oromia, Ethiopia. The study period was from May 1, 2023, to June 30, 2024. Blood glucose levels were examined at five independent intervals: baseline (before vaccination) and follow-up visits at three months, six months, nine months and one-year post-vaccine.

### 2.2. Study population

The study population comprised adult patients diagnosed with type 2 diabetes mellitus (T2DM) based on established clinical criteria. All participants were categorized at the study baseline into two groups: individuals who had received the Johnson & Johnson COVID-19 vaccine and those who had not received any COVID-19 vaccination. This grouping was based solely on participants' vaccination status at the time of enrollment.

### 2.3. Inclusion and exclusion criteria

Participants eligible for the study were adults aged 18 years and above with a confirmed diagnosis of type 2 diabetes mellitus (T2DM) who were actively receiving diabetes treatment. Exclusion criteria included the presence of systemic diseases alongside diabetes, a diagnosis of type 1 diabetes mellitus, a positive test for SARS-CoV-2 infection, or pregnancy complicated by diabetes. Additionally, individuals with a history of severe adverse reactions to vaccines or those with comorbid conditions impacting immune function were also excluded from the study.

### 2.4. Sample size determination

The sample size for this study was determined by accounting for its longitudinal design, which involves repeated measurements from each participant over time. The calculation followed the formula outlined by Diggle and Kenward [19]. Five data collection points were scheduled: baseline, three months, six months, nine months, and one year. Key parameters

used in the sample size calculation included the number of time points (t = 5), a type I error rate (α) of 0.05, a statistical power of 90%, the smallest clinically meaningful difference (d) of 0.5, and an allocation ratio (λ) of 1:3 between vaccinated and unvaccinated groups. The standard deviation (σ) was assumed to be 1, and the correlation between repeated measures (r) was set at 0.02. The imbalance in group sizes was intentional for several reasons: the expected effect size was relatively small (a mean difference of 0.5), and having a larger unvaccinated control group improved the precision of estimates and enhanced statistical power. Moreover, during the initial COVID-19 vaccination campaign, a greater number of unvaccinated individuals were accessible due to low vaccine coverage.

A minimum sample size of 13 vaccinated and 39 unvaccinated participants was required. To account for potential attrition, a 10% adjustment was added to the minimum required sample sizes; therefore, 15 vaccinated and 45 unvaccinated participants were needed. The formula used for the calculation of sample size is expressed below:

$$nt = ((\lambda + 1) * (Z_{(1-\alpha/2)} + Z_\beta)^2 * (\sigma^2) * (1 + (t - 1) * r)) / (\lambda * d * (\mu1 - \mu2)^2)$$

Specifically, due to continuous counseling efforts and active national vaccination campaigns, we initially enrolled 75 vaccinated and 225 unvaccinated participants who met the inclusion criteria. However, throughout the follow-up period, 18 individuals from the vaccinated group and 58 from the unvaccinated group were lost to follow-up, mainly due to vaccination uptake or other reasons. In the end, 57 vaccinated and 167 unvaccinated participants completed the study, maintaining a sample size that exceeded the minimum required for the research.

## 2.5. Data collection

Baseline demographic and clinical data, such as sex, age, medical history, and lifestyle behaviors (smoking, alcohol use, and khat usage), were gathered via structured interviews and examinations of medical records. Participants who were vaccinated received the Johnson & Johnson COVID-19 vaccine (Ad26.COV2.S) following standard administration protocols. Blood samples were collected at four distinct time points: at baseline (prior to vaccination) and during follow-up visits at three months, six months, and one year after vaccination. Blood collection was undertaken using defined techniques, with samples processed through centrifugation to separate serum or plasma. All samples were kept at −80 °C until analysis to retain their integrity.

## 2.6. Random blood sugar (RBS) and hemoglobin A1C (HbA1c) analysis

The assessment of Random Blood Sugar (RBS) and Hemoglobin A1c (HbA1c) levels was performed following the established methods described by Bhat, et al. [20]. For RBS determination, approximately 2.5 mL of venous blood was drawn into sodium fluoride tubes and centrifuged for five minutes to separate the plasma from whole blood. The plasma was then combined with a glucose diluent and analyzed using the COBAS 6000 chemistry analyzer. For HbA1c measurement, 2 mL of venous blood was collected into ethylenediaminetetraacetic acid (EDTA) tubes, thoroughly mixed, and processed using the same analyzer. All procedures for sample collection, storage, and analysis were conducted according to established protocols and stringent quality control standards at the Adama Public Health Research and Referral Laboratory Center.

## 2.7. Statistical analysis

Data analysis was conducted using SPSS version 26.0 and STATA version 18.0. Descriptive statistics were used to summarize baseline characteristics such as age, sex, and medication use. Chi-square tests assessed differences in categorical variables, while independent t-tests evaluated variations in continuous variables between groups. Changes in glycemic outcomes over time were analyzed using Generalized Estimating Equations (GEE) to account for repeated measurements and correlations within subjects. To assess the clinical relevance of observed changes, Cohen's d effect sizes were calculated. Statistical significance was set at a p-value of less than 0.05.

## 2.8. Ethical considerations

Ethical approval for the study was granted by the Institutional Review Board of Addis Ababa University, College of Health Sciences (Protocol No: 019/23/biochemistry), as well as the National Ethics Review Committee of the Ministry of Education (Ref No: 17/152/235/24). Written informed consent was obtained from all participants following comprehensive explanations of the study's objectives, procedures, potential risks, and benefits. The research was conducted in strict accordance with the ethical principles outlined in the Declaration of Helsinki, with particular attention to respecting participant autonomy, promoting beneficence, and ensuring justice [21].

## 3. Results

### 3.1. Participant baseline characteristics stratified by vaccination status

The baseline characteristics of the 224 study participants, categorized by vaccination status, included 57 individuals (25.5%) who were vaccinated and 167 (74.5%) who remained unvaccinated (Table 1). The average age of participants was $43.2 \pm 12.6$ years. Among those aged 40 years or younger, 28.4% were vaccinated compared to 71.6% unvaccinated. For participants over 40 years old, 23.8% were vaccinated while 76.2% were unvaccinated. No statistically significant association was found between age groups and vaccination status ($p = 0.446$, $\chi^2 = 0.580$). Regarding sex distribution, vaccination rates were similar, with 25.2% of females and 25.7% of males vaccinated, showing no significant difference between sexes ($p = 0.936$, $\chi^2 = 0.019$). Likewise, the type of diabetic medication—whether insulin, metformin, or a combination—did not significantly influence vaccination status; 23.2% of insulin users, 22.9% of those on metformin, and 31.8% of participants taking both medications were vaccinated ($p = 0.399$, $\chi^2 = 0.070$). These findings suggest that demographic factors such as age, sex, and medication type were not significantly associated with vaccination status.

### 3.2. Independent test for vaccinated against unvaccinated groups

The independent test comparing vaccinated and unvaccinated groups, as reported in Table 2, demonstrated significant differences in RBS and HbA1c levels across the study intervals. At baseline, no significant difference in RBS levels was observed between the groups (MD = 1.16, 95% CI: −2.54 to 4.86, $p = 0.539$, ES = 0.09). However, significant differences emerged at 3 months post-vaccination (MD = 10.03, 95% CI: 7.88 to 12.19, $p = 0.001$, ES = 0.79) and persisted at 6 months (MD = 3.09, 95% CI: 0.82 to 5.36, $p = 0.008$, ES = 0.24), 9 months (MD = 3.47, 95% CI: 1.22 to 5.73, $p = 0.003$, ES = 0.27), and 1 year (MD = 2.47, 95% CI: 0.22 to 4.73, $p = 0.032$, ES = 0.19). These findings imply that immunization was related with increased RBS levels over time, with the largest effects seen at three months. For HbA1c values, significant differences were seen at all time points, beginning at baseline (MD = 0.24, 95% CI: 0.04 to 0.45, $p = 0.021$, ES = 0.34). The differences

**Table 1. Comparison of baseline demographics and clinical characteristics by vaccination status (N = 224).**

| Variables | Categories | Number of Vaccinated | Number of Unvaccinated | Percentages (%) Vaccinated (n = 57) | Percentages (%) Unvaccinated (n = 167) | p-value | $\chi^2$ |
|---|---|---|---|---|---|---|---|
| Age | ≤ 40 | 23 | 58 | 28.40 | 71.60 | 0.446 | 0.580 |
| | > 40 | 34 | 109 | 23.80 | 76.20 | | |
| | Total | 57 | 167 | 25.40 | 74.60 | | |
| | Mean | 43.2 ± 12.6 | | | | | |
| Sex | Female | 29 | 86 | 25.20 | 74.80 | 0.936 | 0.019 |
| | Male | 28 | 81 | 25.70 | 74.30 | | |
| Medications | Insulin | 13 | 43 | 23.20 | 76.80 | 0.399 | 0.070 |
| | Metformin | 24 | 105 | 22.90 | 77.10 | | |
| | Both | 20 | 63 | 31.80 | 68.30 | | |

**Table 2. Independent test comparing vaccinated and unvaccinated groups across outcome variables at baseline and follow-up intervals.**

| Outcomes | Baseline | | Three months | | Six months | | Nine months | | One year | |
|---|---|---|---|---|---|---|---|---|---|---|
| | MD | ES | MD | ES | MD | ES | MD | ES | MD | ES |
| RBS | 1.16 (−2.54, 4.86) | 0.09 | 10.03 (7.88, 12.19) | 0.79 | 3.09 (0.82, 5.36) | 0.24 | 3.47 (1.22, 5.73) | 0.27 | 2.47 (0.22, 4.73) | 0.19 |
| p value | 0.539 | | 0.001 | | 0.008 | | 0.003 | | 0.032 | |
| HbA1C | 0.24 (0.04, 0.45) | 0.34 | 0.45 (0.33, 0.57) | 0.62 | 0.66 (0.53, 0.79) | 0.91 | 0.89 (0.76, 1.02) | 1.23 | 0.89 (0.76, 1.02) | 1.23 |
| p value | 0.021 | | 0.001 | | 0.001 | | 0.001 | | 0.001 | |

Note: Values were represented as mean difference (95% confidence interval (CI)). MD = Mean difference, ES = Effect Size, RBS = Random Blood Sugar, HbA1C = Haemoglobin A1C

progressively increased at 3 months (MD = 0.45, 95% CI: 0.33 to 0.57, p = 0.001, ES = 0.62), 6 months (MD = 0.66, 95% CI: 0.53 to 0.79, p = 0.001, ES = 0.91), and 9 months (MD = 0.89, 95% CI: 0.76 to 1.02, p = 0.001, ES = 1.23), and remained stable at 1 year (MD = 0.89, 95% CI: 0.76 to 1.02, p = 0.001, ES = 1.23). The significant effect sizes seen over the follow-up period indicate the strong and constant relationship between vaccination and improved glycemic management, with particularly notable improvements in HbA1c levels among the vaccinated group.

### 3.3. Trends in Random Blood Sugar (RBS) levels

Fig 1 illustrates the trends in Random Blood Sugar (RBS) levels for both vaccinated and unvaccinated groups throughout the study period. At baseline, RBS levels were similar between the two groups. However, at three months post-vaccination, the vaccinated group showed a significant rise in RBS levels, reaching a peak at this time point, while the unvaccinated group's RBS levels remained relatively stable. After this peak, RBS levels in the vaccinated group steadily

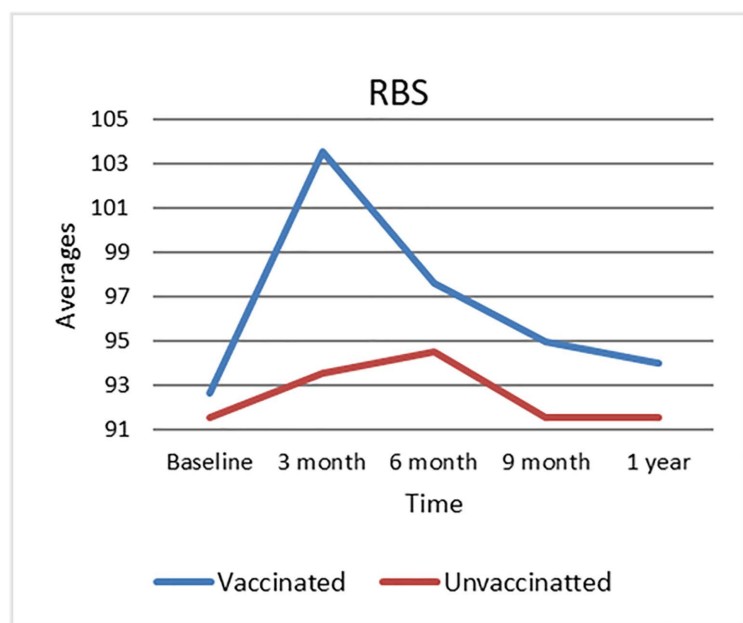

**Fig 1. Random Blood Sugar (RBS) levels over time by vaccination status.** RBS levels (mg/dL) over time in vaccinated and unvaccinated groups. Data points represent group means. Asterisks (*) indicate statistically significant differences between groups at the corresponding time points (p < 0.05), while 'ns' denotes no significant difference.

declined at six months, nine months, and one year, eventually approaching their baseline measurements. In contrast, the unvaccinated group maintained consistent RBS levels with minor variations throughout the study period. These findings demonstrate a temporary spike and subsequent leveling of RBS levels in vaccinated subjects, contrasting with the constant trend found in the unvaccinated group, demonstrating a dynamic response related with vaccination status.

### 3.4. Effect of vaccination on RBS levels

Fig 2 presents the effect size of Random Blood Sugar (RBS) levels in the vaccinated group relative to the unvaccinated group across five time points: baseline, three months, six months, nine months, and one-year post-vaccination. At baseline, the effect size was close to zero, indicating little difference between the two groups. However, the effect size increased sharply at three months post-vaccination, reflecting a substantial impact of vaccination on RBS levels. This effect reduced substantially at six months and continued to decrease, stabilizing with modest fluctuations at nine months and one year. These findings imply that the influence of immunization on RBS levels was temporary, with the most notable effect detected at three months, followed by increasing normalization over time.

### 3.5. Multiple regression analysis of Random Blood Sugar (RBS) using Generalized Estimating Equations (GEE)

The Generalized Estimating Equations (GEE) multiple regression analysis in Table 3 identifies key parameters determining RBS levels. Vaccination status was substantially linked with elevated RBS levels, as vaccinated subjects showed significant increases in both crude ($\beta = 4.04$; 95% CI: 2.01, 6.08; $p < 0.001$) and adjusted models ($\beta = 4.81$; 95% CI: 2.79, 6.83; $p < 0.001$). Time was also a crucial factor, with the most substantial increases occurring at three months (adjusted $\beta = 4.26$; 95% CI: 3.34, 5.18; $p < 0.001$) and six months (adjusted $\beta = 3.49$; 95% CI: 2.71, 4.27; $p < 0.001$), followed by non-significant differences at nine months and one year, indicating a declining effect over time. Younger participants (aged ≤ 40 years) displayed greater RBS levels (adjusted $\beta = 14.33$; 95% CI: 12.20, 16.47; $p < 0.001$) compared to older participants, while females had higher RBS levels than males (adjusted $\beta = 6.15$; 95% CI: 3.92, 8.37; $p < 0.001$). Lifestyle factors such as alcohol, khat, and smoking exposure revealed no significant association with RBS levels, whereas the impacts of drugs like metformin or insulin were minor, with metformin's relevance decreasing after adjustment. These findings indicate the considerable influence of vaccination, time, age, and sex on RBS levels, with the largest effects reported within the first six months post-vaccine.

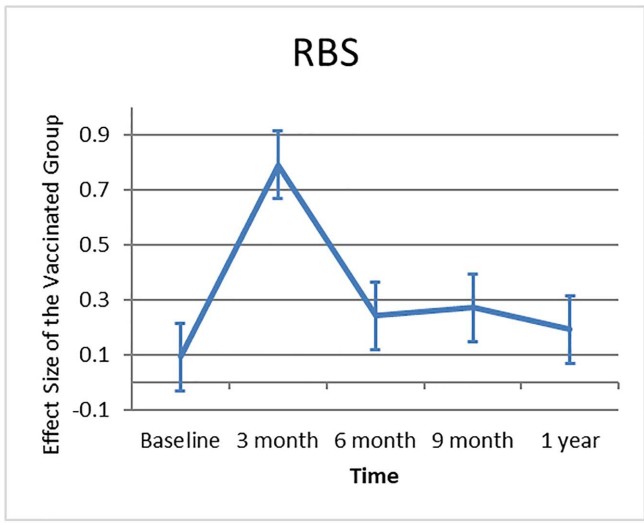

**Fig 2. Effect size of RBS levels in the vaccinated group over time.**

Table 3. GEE multiple regression of Random Blood Sugar (RBS) crude and adjusted effects.

| Variables | Categories | Crude Effect | | Adjusted Effect | |
|---|---|---|---|---|---|
| | | β (95% CI) | p value | β (95% CI) | p value |
| Vaccination Status | Vaccinated | 4.04 (2.01, 6.08) | < 0.001 | 4.81 (2.79, 6.83) | < 0.001 |
| | Unvaccinated | Reference | | Reference | |
| Time | | | < 0.001 | | < 0.001 |
| | Three months | 4.26 (3.34, 5.18) | < 0.001 | 4.26 (3.34, 5.18) | < 0.001 |
| | Six months | 3.49 (2.71, 4.27) | < 0.001 | 3.49 (2.71, 4.27) | < 0.001 |
| | Nine months | 0.59 (.17, 1.35) | 0.128 | 0.59 (.17, 1.35) | 0.128 |
| | One-year | 0.33 (.42, 1.09) | 0.382 | 0.33 (.42, 1.09) | 0.382 |
| | Base line | Reference | | Reference | |
| Age group | ≤ 40 years | 13.81 (16.12, 11.49) | < 0.001 | 14.33 (16.47, 12.20) | < 0.001 |
| | > 40 years | Reference | | Reference | |
| Sex | Female | 5.51 (8.39, 2.62) | < 0.001 | 6.15 (8.37, 3.92) | < 0.001 |
| | Male | Reference | | Reference | |
| Alcohol exposure | Yes | 0.02 (3.36, 3.39) | 0.992 | | |
| | No | Reference | | | |
| Khat exposure | Yes | 0.57 (3.83, 4.97) | 0.799 | | |
| | No | Reference | | | |
| Smoking exposure | Yes | 1.28 (4.26, 6.81) | 0.651 | | |
| | No | Reference | | | |
| Medications | | | 0.001 | | 0.245 |
| | Insulin | 1.58 (2.16, 5.32) | 0.408 | 1.36 (1.34, 4.06) | 0.323 |
| | Metformin | 5.78 (2.69, 8.87) | <.001 | 1.83 (.52, 4.18) | 0.127 |
| | Both | Reference | | Reference | |
| Intercept | | | | 98.94 | |

Note: N = 224; p value ≤ 0.20; p value < 0.05. The "Reference" category represents the group used for comparison in the regression model. This approach facilitates understanding of how vaccination status, time, and other factors influence glycemic control over the study period.

### 3.6. Trends in Hemoglobin A1C (HbA1C) levels

Fig 3 demonstrates the trends in hemoglobin A1C (HbA1C) levels over time, stratified by vaccination status. At baseline, both vaccinated and unvaccinated groups start with comparable HbA1C values, demonstrating similar glycemic management prior to the trial. In the vaccinated group, HbA1C values indicate a steady increase, peaking at 6 months before modestly dropping, while maintaining above baseline levels at the one-year point. In contrast, the unvaccinated group demonstrates a steady downward trend, with HbA1C levels progressively dropping over time and reaching their lowest point at one year. These divergent trajectories reflect a divergence in glycemic control across the groups, with the vaccinated group displaying decreasing metabolic regulation over time, while the unvaccinated group displays consistent improvement throughout the study period.

### 3.7. Effect sizes of RBS and HbA1C levels over time in the vaccinated group

The effect sizes of RBS and HbA1C levels over time in the vaccinated group, as shown in Fig 4, indicate significant trends. At baseline, the effect sizes for both RBS and HbA1C are around zero, indicating no substantial departures from the reference point. By the 3-month mark, both measures increase, with HbA1C exhibiting a bigger impact magnitude than RBS. RBS impact size peaks at 6 months and gradually drops, staying positive but

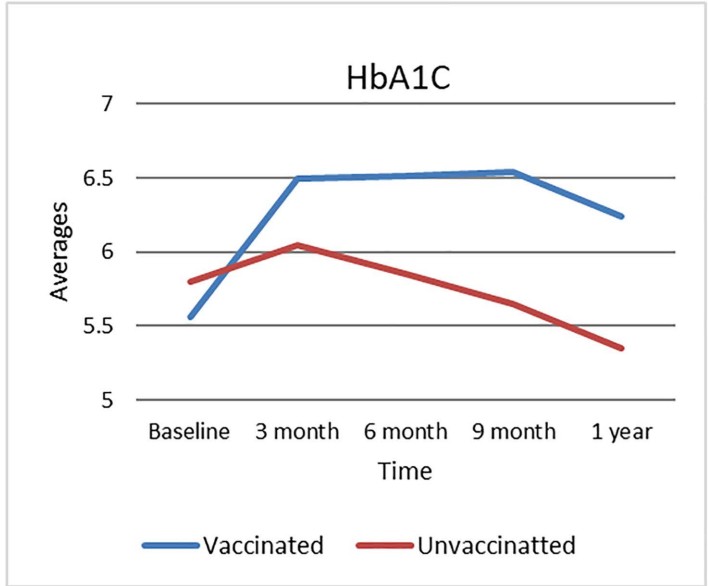

**Fig 3. HbA1C levels over time by vaccination status.** HbA1C levels (mg/dL) over time in vaccinated and unvaccinated groups. Data points represent group means. Asterisks (*) indicate statistically significant differences between groups at the corresponding time points (p<0.05), while 'ns' denotes no significant difference.

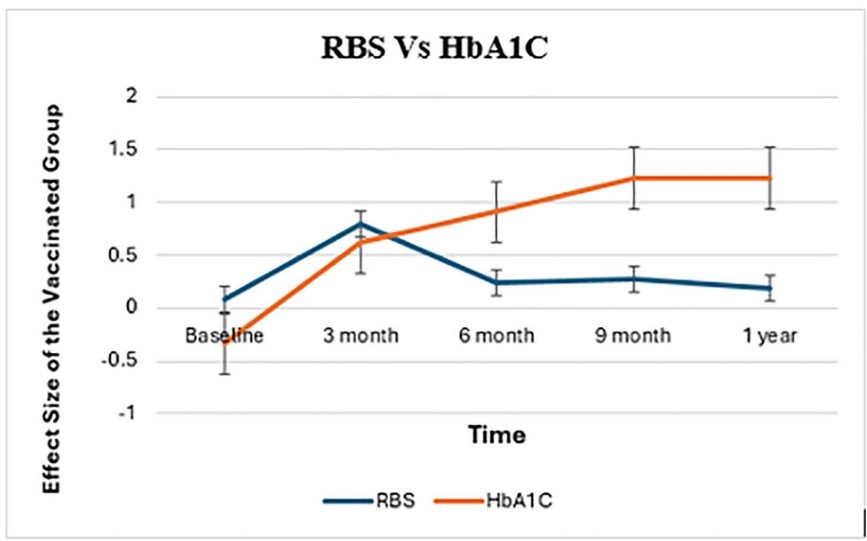

**Fig 4. Effect size of RBS and HbA1C levels over time by vaccination status.**

decreasing at 9 months and one year. In contrast, HbA1C exhibits a continuous increasing trajectory, reaching its peak impact size at the one-year mark. These data demonstrate that while the influence of immunization on RBS is temporary, HbA1C levels are progressively and permanently impacted, demonstrating a long-term shift in glycemic management.

### 3.8. Multiple regression analysis of hemoglobin A1c (HbA1c) using Generalized Estimating Equations (GEE)

Table 4 presents the Generalized Estimating Equations (GEE) analysis identifying key factors influencing HbA1c levels. Vaccination status was significantly associated with increased HbA1c, as the vaccinated group showed a crude effect size of β = 0.53 (95% CI: 0.43, 0.63; p < 0.001) and an adjusted effect of β = 0.23 (95% CI: 0.39, 0.07; p = 0.005) compared to the unvaccinated group. Time also emerged as a significant predictor: HbA1c levels peaked at three months post-vaccination (adjusted β = 0.24; 95% CI: 0.11, 0.38; p < 0.001), slightly declined at six months (adjusted β = 0.04; 95% CI: −0.09, 0.18; p = 0.512), and then significantly decreased at nine months (adjusted β = −0.16; 95% CI: −0.29, −0.02; p = 0.024) and one year (adjusted β = −0.46; 95% CI: −0.59, −0.32; p < 0.001) relative to baseline. Interaction effects between vaccination and time revealed consistently higher HbA1c levels in the vaccinated group across all time points, with the most pronounced differences at nine months and one year (both adjusted β = 1.13; 95% CI: 0.92, 1.34; p < 0.001).

**Table 4. GEE multiple regression of Hemoglobin A1C (HbA1C) crude and adjusted effects.**

| Variables (N = 224) | Categories | Crude Effect | | Adjusted Effect | |
|---|---|---|---|---|---|
| | | β (95% CI) | p value | β (95% CI) | p value |
| Vaccination Status | Vaccinated | 0.53 (0.43, 0.63) | < 0.001 | 0.23 (0.39, 0.07) | 0.005 |
| | Unvaccinated | Reference | | Reference | |
| Time | | | < 0.001 | | < 0.001 |
| | Three months | 0.42 (0.31, 0.54) | < 0.001 | 0.24 (0.11, 0.38) | < 0.001 |
| | Six months | 0.28 (0.15, 0.39) | < 0.001 | 0.04 (−0.09, 0.18) | 0.512 |
| | Nine months | 0.13 (0.01, 0.26) | 0.039 | −0.16 (0.29, −0.02) | 0.024 |
| | One-year | 0.17 (0.29, 0.04) | 9 | −0.46 (0.59, −0.32) | < 0.001 |
| | Base line | Reference | | Reference | |
| Vaccinated group * Time | | | | | < 0.001 |
| | Vaccinated * 3 month | | | 0.69 (0.48, 0.90) | < 0.001 |
| | Vaccinated * 6 month | | | 0.91 (0.69, 1.12) | < 0.001 |
| | Vaccinated * 9 month | | | 1.13 (0.92, 1.34) | < 0.001 |
| | Vaccinated * 1 year | | | 1.13 (0.92, 1.34) | < 0.001 |
| | Vaccinated * Baseline | | | Reference | |
| Age group | ≤ 40 years | 0.24 (0.39, 0.08) | 0.003 | 0.27 (0.41, 0.14) | < 0.001 |
| | > 40 years | Reference | | Reference | |
| Sex | Female | 0.21 (0.37, 0.06) | 0.006 | 0.22 (0.36, 0.09) | 0.001 |
| | Male | Reference | | Reference | |
| Alcohol exposure | Yes | 0.00 (0.17, 0.17) | 0.993 | | |
| | No | Reference | | | |
| Khat exposure | Yes | 0.11 (0.15, 0.38) | 0.392 | | |
| | No | Reference | | | |
| Smoking exposure | Yes | 0.12 (0.27, 0.51) | 0.556 | | |
| | No | Reference | | | |
| Medications | | | 0.391 | | |
| | Insulin | 0.09 (0.30, 0.13) | 0.436 | | |
| | Metformin | 0.05 (0.15, 0.24) | 0.642 | | |
| | Both | Reference | | | |
| Intercept | | | | 6.01 | |

Note: N = 224; p value ≤ 0.20; p value < 0.05. The "Reference" category represents the group used for comparison in the regression model. This approach facilitates understanding of how vaccination status, time, and other factors influence glycemic control over the study period.

Age was also a significant factor, with younger individuals (≤ 40 years) exhibiting lower HbA1c levels (adjusted $\beta = -0.27$; 95% CI: −0.41, −0.14; $p < 0.001$) compared to older participants. Furthermore, females had significantly lower HbA1c levels than males (adjusted $\beta = -0.22$; 95% CI: −0.36, −0.09; $p = 0.001$). Lifestyle factors, including alcohol use, khat exposure, and smoking, showed no significant relationships with HbA1C, while medicines such as metformin or insulin similarly demonstrated low influence. These data underscore that vaccination, time, age, and sex were the key predictors of HbA1C levels, with unique group-time trends identified in the vaccinated population.

## 4. Discussion

The study investigated the impact of the Johnson & Johnson (Ad26.COV2.S) COVID-19 vaccine on glycemic control among individuals diagnosed with T2DM. Key findings revealed crucial demographic variances, trends in glycemic parameters, and factors affecting glycemic variability, providing insights into the metabolic effects of viral vector-based vaccines in diabetic populations.

Demographic analysis indicated no significant correlation between vaccination status and age or sex, aligning with studies that suggest vaccine uptake is generally widespread across different demographic groups [22]. However, younger individuals exhibited greater variability in Random Blood Sugar (RBS) levels post-vaccination, consistent with evidence that younger people generate stronger immune responses, which may accentuate metabolic fluctuations [23]. Similarly, females experienced more pronounced hyperglycemic alterations than males, potentially due to sex-based differences in immune reactivity and hormonal influences. Prior research confirms that women often mount heightened immune responses, characterized by greater cytokine production following vaccination, which can exacerbate insulin resistance and glucose dysregulation [24,25].

The transient increase in RBS levels shortly after receiving the Ad26.COV2.S vaccine aligns with previous findings on brief glycemic disturbances following COVID-19 immunization [26]. Viral vector vaccines like Ad26.COV2.S trigger immune activation by releasing pro-inflammatory cytokines, notably interleukin-6 (IL-6) and tumor necrosis factor-alpha (TNF-α). These cytokines are recognized for impairing insulin sensitivity and promoting glucose production in the liver, which can lead to temporary elevations in blood sugar levels [27–29]. Research involving COVID-19 patients indicates that systemic inflammation—especially the surge of cytokines such as IL-6 and TNF-α—significantly contributes to insulin resistance and the development of hyperglycemia [30]. This inflammatory response is a well-documented mechanism in COVID-19 itself and can similarly be triggered by vaccination, leading to transient increases in RBS and fluctuations in Hemoglobin A1c (HbA1c) levels [31]. Nevertheless, the eventual stabilization of RBS levels highlights the temporary nature of these metabolic disturbances, reinforcing the vaccine's overall safety in the short term. Similar glycemic fluctuations have been observed with other COVID-19 vaccines, with glucose homeostasis typically restoring as immune activation declines [32].

However, Hemoglobin A1c (HbA1c) levels showed a gradual increase over time in the vaccinated group, suggesting potential long-term metabolic consequences. This finding is consistent with research indicating that post-vaccination HbA1c elevations in diabetic patients may result from recurrent episodes of transient hyperglycemia or persistent low-grade inflammation [33,17]. Chronic immune activation, even at subclinical levels, can worsen insulin resistance and impair glucose regulation [34]. Moreover, stress-induced hyperglycemia has been documented in hospitalized COVID-19 patients, attributed to activation of the hypothalamic-pituitary-adrenal (HPA) axis. This activation elevates cortisol secretion, which disrupts normal glucose metabolism [30]. A similar mechanism may contribute to post-vaccination hyperglycemia, particularly in individuals with preexisting diabetes, as the vaccine-induced immune response places metabolic stress on the body. This aligns with findings from hematological and biochemical studies, which show that acute-phase responses, including increased blood glucose levels, occur following immune activation [35]. On the other hand, some studies report no significant HbA1c elevations in well-controlled diabetes patients, emphasizing the importance of baseline metabolic status and individual variability in determining vaccine-related outcomes [36].

                                                                                

Predictors of glycemic variability in this study included vaccination status, time since immunization, age, and sex. Vaccination was strongly linked to elevated RBS levels, aligning with findings that COVID-19 vaccine-induced immune responses can disrupt glucose control [29]. The highest glycemic fluctuations occurred immediately post-vaccination and declined over time, underscoring the dynamic interplay between vaccine-induced inflammation and glucose metabolism [27]. Age and sex were also key determinants, with younger individuals and females displaying greater glycemic variability, supporting evidence that immune and metabolic responses vary across demographic groups [23,27]. Furthermore, individuals with poor glycemic control before infection or vaccination tend to experience more pronounced fluctuations in blood glucose levels. Higher baseline HbA1c levels have been associated with worsened COVID-19 outcomes, as they contribute to greater inflammation and oxidative stress, further impairing glucose regulation [30,31]. These findings highlight the importance of monitoring glycemic changes post-vaccination, particularly in populations with high diabetes prevalence.

The role of diabetes medications in mitigating vaccine-induced glycemic changes appeared limited. Although metformin has been associated with improved glycemic stability and reduced COVID-19 severity, its impact on post-vaccination glycemic alterations in this study was minimal [37]. Similarly, insulin usage did not significantly affect glycemic variability, suggesting that observed metabolic disturbances were primarily driven by immunization status and demographic or clinical factors. Additionally, lifestyle factors such as alcohol consumption and smoking showed no significant influence on glycemic outcomes, reinforcing the notion that vaccine-induced immune responses are the predominant drivers of metabolic changes [38].

Despite these glycemic fluctuations, the Johnson & Johnson (Ad26.COV2.S) vaccine remains a crucial intervention for reducing severe COVID-19 outcomes in diabetic populations. Individuals with T2DM face a markedly increased risk of severe complications—such as hospitalization and death—primarily because of their chronic pro-inflammatory condition and related comorbidities [33,17]. While the transient nature of RBS disturbances and the gradual yet manageable increase in HbA1c support the vaccine's overall safety, the implications of sustained metabolic changes warrant further consideration. Studies have shown that viral vector-based vaccines trigger inflammatory pathways similar to those seen in post-COVID metabolic syndrome, where persistent cytokine activation contributes to long-term cardiovascular and metabolic dysfunction [39]. This suggests that for certain individuals, particularly those with metabolic syndrome or obesity, vaccine-induced inflammation may lead to prolonged metabolic disturbances. However, conflicting evidence exists, with some studies indicating no substantial long-term HbA1c changes in well-controlled diabetes patients, underscoring the role of individual metabolic variability [27]. Tudoran, et al. [40] further highlight those individuals with metabolic syndrome and obesity experience exacerbated inflammatory responses post-COVID, raising concerns that a similar prolonged immune activation post-vaccination might underlie glycemic variability. Nevertheless, as most hyperglycemic alterations resolve as immune activity subsides [32], the benefits of COVID-19 vaccination continue to outweigh the potential metabolic risks. These findings reinforce the necessity of integrating vaccination campaigns with diabetes management strategies, particularly in high-risk populations, to optimize glycemic outcomes while maintaining robust immunization coverage.

### 4.1. Strengths and limitations

However, several limitations should be acknowledged. First, the study was conducted at a single center—Adama Hospital Medical College in Ethiopia—which may limit the external validity and generalizability of the findings to other populations with different demographic or healthcare characteristics. Although adjustments were made for key confounders, residual confounding from unmeasured variables—such as dietary patterns, physical activity, psychosocial stress, and socioeconomic status—could have influenced the glycemic outcomes.

Moreover, despite statistical adjustments to account for differences between the vaccinated and unvaccinated groups, the potential for selection bias remains. This is particularly relevant given the non-randomized design and unequal group sizes. Future research would benefit from implementing propensity score matching (PSM) to improve comparability by

balancing observed covariates across groups. Additionally, conducting more comprehensive sensitivity analyses would help assess the robustness of the findings.

Although the current analytical framework using Generalized Estimating Equations (GEE) is statistically sound, the incorporation of advanced techniques—such as PSM, mixed-effects modeling, or instrumental variable analysis—could further strengthen the methodological rigor of future studies. Finally, larger multi-center investigations are recommended to validate these findings and provide broader insights into the metabolic effects of COVID-19 vaccination in individuals with T2DM. In addition to measuring RBS, it would have been beneficial to include glucose monitoring; however, we could not implement this due to resource limitations.

## 5. Conclusion

This study demonstrates that the Johnson & Johnson COVID-19 vaccine is associated with transient increases in random blood sugar and a sustained elevation in hemoglobin A1c levels among individuals with Type 2 Diabetes Mellitus. While the short-term glycemic fluctuations observed post-vaccination suggest an inflammatory or metabolic response, the gradual rise in HbA1c underscores the need for continuous glycemic monitoring in vaccinated individuals. These findings highlight the interplay between immune activation and glucose metabolism, emphasizing the importance of personalized diabetes management following vaccination. Despite these metabolic changes, the vaccine's substantial protective benefits against severe COVID-19 far outweigh the temporary disturbances in glycemic control. Integrating vaccination efforts with tailored diabetes care strategies is essential to optimizing outcomes in high-risk populations. Future research should focus on elucidating the underlying mechanisms driving these metabolic variations and exploring interventions to mitigate long-term glycemic effects.

## Supporting information

**S1 File. F1- Glycemic data used for analysis.**
(XLSX)

## Acknowledgments

The authors express their sincere appreciation to the staff of Adama Hospital Medical College for their dedicated support throughout the study. Special thanks are also extended to all participants for their valuable contributions and commitment to the research process.

## Author contributions

**Conceptualization:** Chala Kenenisa Edae, Abdisa Tufa Bedada, Maria Degef Teklemariam, Mesfin Bekele Tolesa, Solomon Genet Gebre.

**Data curation:** Chala Kenenisa Edae, Abdisa Tufa Bedada, Maria Degef Teklemariam, Mesfin Bekele Tolesa, Solomon Genet Gebre.

**Formal analysis:** Chala Kenenisa Edae, Abdisa Tufa Bedada, Maria Degef Teklemariam, Mesfin Bekele Tolesa, Solomon Genet Gebre.

**Funding acquisition:** Chala Kenenisa Edae, Mesfin Bekele Tolesa.

**Investigation:** Chala Kenenisa Edae, Maria Degef Teklemariam, Mesfin Bekele Tolesa, Solomon Genet Gebre.

**Methodology:** Chala Kenenisa Edae, Abdisa Tufa Bedada, Maria Degef Teklemariam, Mesfin Bekele Tolesa, Solomon Genet Gebre.

**Project administration:** Chala Kenenisa Edae, Abdisa Tufa Bedada, Solomon Genet Gebre.

**Resources:** Chala Kenenisa Edae, Abdisa Tufa Bedada, Solomon Genet Gebre.

**Software:** Chala Kenenisa Edae.

**Supervision:** Chala Kenenisa Edae, Abdisa Tufa Bedada, Maria Degef Teklemariam, Solomon Genet Gebre.

**Validation:** Chala Kenenisa Edae, Abdisa Tufa Bedada, Maria Degef Teklemariam, Mesfin Bekele Tolesa, Solomon Genet Gebre.

**Visualization:** Chala Kenenisa Edae, Abdisa Tufa Bedada.

**Writing – original draft:** Chala Kenenisa Edae, Abdisa Tufa Bedada, Maria Degef Teklemariam, Mesfin Bekele Tolesa, Solomon Genet Gebre.

**Writing – review & editing:** Chala Kenenisa Edae, Abdisa Tufa Bedada, Maria Degef Teklemariam, Mesfin Bekele Tolesa, Solomon Genet Gebre.

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
