## [Decision Letter · Decision Letter 0]

16 Sep 2025

Dear Dr. Edae,

Thank you for submitting your manuscript to PLOS ONE. After careful consideration, we feel that it has merit but does not fully meet PLOS ONE’s publication criteria as it currently stands. Therefore, we invite you to submit a revised version of the manuscript that addresses the points raised during the review process.

We look forward to receiving your revised manuscript.

Kind regards,

Timotius Ivan Hariyanto, M.D.

Academic Editor

PLOS ONE

Journal Requirements:

“This study was financially supported by Addis Ababa University and the Oromia Health Bureau.”

4. In the online submission form, you indicated that “The data supporting the findings of this study are available from the authors upon reasonable request.”

Reviewers' comments:

Reviewer's Responses to Questions

**Comments to the Author**

1. Is the manuscript technically sound, and do the data support the conclusions?

Reviewer #1: Partly

Reviewer #2: Partly

2. Has the statistical analysis been performed appropriately and rigorously?

Reviewer #1: No

Reviewer #2: Yes

3. Have the authors made all data underlying the findings in their manuscript fully available?

Reviewer #1: No

Reviewer #2: Yes

4. Is the manuscript presented in an intelligible fashion and written in standard English?

Reviewer #1: No

Reviewer #2: Yes

Reviewer #1: Manuscript ID: PONE-D-25-35535

Title: Prospective Assessment of Johnson & Johnson COVID-19 Vaccine Effects on Glycemic Biomarkers in Type 2 Diabetes Mellitus in Ethiopia

General Evaluation

This study investigates the effects of the Johnson & Johnson COVID-19 vaccine (Ad26.COV2.S) on glycemic biomarkers among patients with type 2 diabetes mellitus (T2DM) in Ethiopia. The authors conducted a prospective longitudinal study to compare glycemic trends between vaccinated and unvaccinated individuals with diabetes. The research is timely, relevant, and well-structured; however, there are several significant and minor issues that need to be addressed to enhance scientific rigor, interpretation, and contextualization.

Major Comments

Insufficient Prior Research on Glycemic Control and Vaccine Response

The manuscript does not adequately reference key studies that demonstrate the direct relationship between glycemic control and the immunogenicity and efficacy of COVID-19 vaccines, particularly in patients with T2DM. Essential studies include:

Marfella et al., Nat Commun (2022): This research indicates that poor glycemic control is associated with increased SARS-CoV-2 breakthrough infections post-vaccination among T2DM patients.

Sardu et al., J Clin Med (2022): This study explores the pathophysiological mechanisms through which hyperglycemia may impair vaccine effectiveness and worsen illness severity.

Marfella et al., Diabetes Obes Metab (2022): The CAVEAT Study examines the impact of glycemic control on the immunogenicity of COVID-19 vaccines in T2DM patients.

Inclusion of these references in the Introduction and Discussion sections is crucial for accurately situating your findings within the existing body of literature.

Clarification of HbA1c Trends

The assertion that vaccination resulted in a “sustained increase” in HbA1c should be articulated with greater precision. Due to the non-randomized study design, it is difficult to eliminate the influence of baseline confounders or lifestyle changes. Although the Generalized Estimating Equations (GEE) model accounts for several covariates, it does not control for unmeasured confounders (e.g., dietary habits or exercise patterns).

Control Group Assignment and Causal Inference

As this study is a prospective cohort analysis, the control group (those not vaccinated) is self-selected, which introduces potential bias. Even retrospectively, methods such as Propensity Score Matching (PSM) or Inverse Probability Weighting (IPW) could be implemented to adjust for differences between the two groups.

Absence of Immunological Markers

While inflammatory cytokines (IL-6, TNF-α) are mentioned as mechanistic mediators, they are not measured in this study. This limitation should be explicitly acknowledged in the manuscript’s mechanistic interpretations.

Terminology: “Improved” vs. “Increased”

In multiple instances (e.g., Table 2 and the Discussion), the authors use the term “improved” to describe higher values of RBS or HbA1c. This terminology is misleading; it would be more accurate to state that these values are elevated, which typically indicates poorer glycemic control.

Minor Comments

Effect Size Clarification

While using Cohen’s d is appropriate, it is important to contextualize this statistic in terms of clinical significance (e.g., what does an increase of 0.89% in HbA1c imply for patient risk?).

Ethics Statement

The ethics statement is well-written and adheres to the required guidelines.

Figures

All figures should be incorporated into the main body of the manuscript or supplementary materials for improved readability.

Writing Quality

The manuscript is generally well-composed; however, certain sections, particularly the Results, would benefit from professional copy-editing to enhance fluency.

Recommendation

Major Revision

Suggested Citations for Inclusion

Please consider including and discussing the following pivotal studies:

Marfella R et al. Glycaemic control is related to SARS-CoV-2 breakthrough infections in vaccinated patients with type 2 diabetes. Nat Commun. 2022;13(1):2318. [PMID: 35484164]

Sardu C et al. A summary of the impact of hyperglycemia on COVID-19 outcomes: Vaccines efficacy, disease severity and molecular mechanisms. J Clin Med. 2022;11(6):1564. [PMID: 35329890]

Marfella R et al. Does the immunogenicity of the COVID-19 vaccination decrease in type 2 diabetic patients with poor glycaemic control: The CAVEAT study. Diabetes Obes Metab. 2022;24(1):160–165. [PMID: 34494705]

Reviewer #2: Edae CK et al. investigated the impact of the Johnson & Johnson COVID-19 vaccine on glycemic parameters in patients with type 2 diabetes (T2D) in Ethiopia. They demonstrated that Johnson & Johnson COVID-19 vaccination was associated with short term Random Blood Sugar (RBS) fluctuations and a sustained increase in HbA1c levels, with more pronounced hyperglycemic alterations in younger individuals and female patients. Though the topic is potentially relevant, particularly in relatively understudied populations such as African populations, and the prospective design valuable, several substantial revisions are required before the manuscript can be considered for publication.

Major Comments

-Since this study was conducted in Ethiopia, it is important to discuss how these findings might translate to other contexts, particularly in higher-income countries where healthcare structures, diabetes management practices, and vaccination strategies differ significantly. Please, expand the discussion section relative to these aspects.

-Another key limitation of the study is that glycemic variability was assessed through RBS levels rather than more robust methods such as continuous or flash glucose monitoring systems. Though this is likely related to resource constraints, this limitation should be clearly acknowledged and discussed.

-Several important baseline characteristics of the study population are currently missing. I would strongly recommend providing a new detailed table (new Table 1) including for instance data on BMI, smoking status, other comorbidities such as prevalence of hypertension and diabetes-related complications, socioeconomic status, education level, baseline HbA1c levels and other relevant biochemical parameters (e.g. renal function tests), if available, and analyzing potential differences between vaccinated and unvaccinated groups at baseline according to these variables

-Potentially important variables were not included in the multiple regression analyses. In particular, BMI and baseline HbA1c may both influence glycemic control and variability after vaccination. If available, these data should be incorporated into the model, otherwise their absence should be acknowledged as an additional limitation.

-The conclusions could be expanded to include more practical clinical implications of the study findings. How should clinicians interpret these changes in glycemic levels post-vaccination? Should closer monitoring be advised in certain subgroups (e.g., younger individuals, women)? Please, better discuss this point.

Minor Comments

1. Please consider adding a study flowchart figure.

2. Table 2 should be reformatted for greater clarity, with a more clearer separation of columns.

3. Please carefully revise the acronyms throughout the manuscript (e.g., Type 2 Diabetes Mellitus/T2DM, Hemoglobin A1c/HbA1c, Random Blood Sugar/RBS, Hypothalamic-Pituitary-Adrenal/HPA axis).

**Do you want your identity to be public for this peer review?** For information about this choice, including consent withdrawal, please see our Privacy Policy

Reviewer #1: **Yes: ** Raffaele Marfella

Reviewer #2: No

---

## [Author Response · Author response to Decision Letter 1]

29 Sep 2025

We sincerely thank you, the Academic Editor, and the reviewers for your valuable time and constructive feedback on our manuscript. We have carefully revised the paper in line with all the points raised. Below, we provide a detailed, point-by-point response. Reviewer comments are restated in italics, followed by our responses and corresponding changes in the manuscript.

Journal Requirements

1. Style requirements and file naming

Comment: Please ensure manuscript follows PLOS ONE style templates.

Response: We reformatted the manuscript according to PLOS ONE’s formatting requirements. File names for resubmission now follow the required conventions:

• “Response to Reviewers”

• “Revised Manuscript with highlight Changes”

• “Manuscript”

2. Funding information mismatch

Comment: The ‘Funding Information’ and ‘Financial Disclosure’ sections do not match.

Response: We corrected this discrepancy. The Funding section now clearly states:

“There is no funding or grant received for this study.”

3. Role of funders

Comment: Please clarify the role of funders.

Response:

“There is no funder for this study and therefore, no roles.”

4. Data availability policy

Comment: All PLOS journals require underlying data to be freely available.

Response:

“All relevant data within the manuscript were uploaded with the title of supplementary files to be freely accessible.”

5. Citing suggested studies

Comment: Reviewers recommended citing additional work.

Response: We incorporated the following references in the study:

• Marfella R et al., Nat Commun 2022.

• Sardu C et al., J Clin Med 2022.

• Marfella R et al., Diabetes Obes Metab 2022.

Reviewer #1

Major Comments

1. Insufficient prior research on glycemic control and vaccine response.

Response: We revised the Introduction and Discussion to integrate the suggested studies and explain how poor glycemic control may impact vaccine immunogenicity and outcomes.

2. Clarification of HbA1c trends.

Response: We rephrased the Results and Discussion to state “elevated HbA1c levels were observed” rather than “sustained increase,” and emphasized potential influence of unmeasured confounders despite the GEE adjustments.

3. Control group assignment bias.

Response: We acknowledged the non-randomized nature of the study and discussed potential bias due to self-selection. Additionally, we included the need to conduct sensitivity analysis using Propensity Score Matching (PSM) to adjust for baseline differences as a limitation of the study.

4. Absence of immunological markers.

Response: We explicitly acknowledged this limitation in the Discussion and clarified that mechanistic interpretations remain speculative.

5. Terminology: “improved” vs. “increased.”

Response: We revised all instances of “improved” to “elevated” or “increased” when describing RBS and HbA1c.

Minor Comments

• Effect size clarification: We added a discussion on the clinical significance of HbA1c changes.

• Figures: All figures are now embedded in the main text and also available as supplementary files.

• Writing quality: We edited the Results section for clarity and fluency.

Reviewer #2

Major Comments

1. Contextualization to other settings.

Response: We expanded the Discussion to compare findings in Ethiopia with potential implications for high-income countries, where healthcare infrastructure and vaccination strategies differ.

2. Limitation: RBS vs. continuous glucose monitoring.

Response: We acknowledged that RBS is a less robust marker than continuous or flash glucose monitoring, but explained this was due to resource limitations in our setting.

3. Expanded clinical implications.

Response: The Conclusion now advises that clinicians may consider closer glycemic monitoring post-vaccination in younger and female T2DM patients, particularly those with poor baseline glycemic control.

Minor Comments

1. Table 2 formatting: Reformatted for clarity with improved column separation.

2. Acronyms: Revised throughout the manuscript for consistency.

Conclusion

We believe that the revised manuscript has significantly improved in scientific rigor, contextualization, and clarity following the reviewers’ helpful comments. We are grateful for the opportunity to resubmit and look forward to your further consideration.

Sincerely,

Chala Kenenisa Edae

---

## [Decision Letter · Decision Letter 1]

15 Dec 2025

Dear Dr. Edae,

Thank you for submitting your manuscript to PLOS ONE. After careful consideration, we feel that it has merit but does not fully meet PLOS ONE’s publication criteria as it currently stands. Therefore, we invite you to submit a revised version of the manuscript that addresses the points raised during the review process.

We look forward to receiving your revised manuscript.

Kind regards,

Timotius Ivan Hariyanto, M.D.

Academic Editor

PLOS One

Journal Requirements:

Reviewers' comments:

Reviewer's Responses to Questions

**Comments to the Author**

Reviewer #1: All comments have been addressed

Reviewer #3: All comments have been addressed

2. Is the manuscript technically sound, and do the data support the conclusions?

Reviewer #1: Yes

Reviewer #3: Yes

3. Has the statistical analysis been performed appropriately and rigorously?

Reviewer #1: Yes

Reviewer #3: Yes

4. Have the authors made all data underlying the findings in their manuscript fully available?

Reviewer #1: Yes

Reviewer #3: Yes

5. Is the manuscript presented in an intelligible fashion and written in standard English?

Reviewer #1: Yes

Reviewer #3: Yes

Reviewer #1: The manuscript has been further improved and refined for clarity and overall quality. Overall, the scientific message is ok

Reviewer #3: Minor Comments

In the Introduction, condense lines 1–3 to avoid repetition of global DM prevalence.

Define all abbreviations (e.g., GEE, PSM) at first mention.

Correct small typos: “uninfected group” → “unvaccinated group.”

Consider adding 95 % CI ribbons or error bars to trend figures for clarity.

**Do you want your identity to be public for this peer review?** For information about this choice, including consent withdrawal, please see our Privacy Policy

Reviewer #1: **Yes: ** Raffaele Marfella

Reviewer #3: **Yes: ** Elabbass Ali Abdelmahmuod

---

## [Author Response · Author response to Decision Letter 2]

16 Dec 2025

Response to Review Comments

We thank the reviewers for their careful review and constructive feedback, which have helped improve the clarity and quality of the manuscript. We address the comments below.

Reviewer #1

Comment:

The manuscript has been further improved and refined for clarity and overall quality. Overall, the scientific message is acceptable.

Response:

We thank the reviewer for the positive assessment of the manuscript and for acknowledging the improvements made. We are pleased that the scientific message is considered sound.

Reviewer #3 – Minor Comments

Comment:

Condense lines 1–3 of the Introduction to avoid repetition of global DM prevalence.

Response:

The Introduction has been revised to condense lines 1–3 and remove repetitive statements on global diabetes mellitus prevalence.

Comment:

Define all abbreviations (e.g., GEE, PSM) at first mention.

Response:

All abbreviations, including GEE (Generalized Estimating Equations) and PSM (Propensity Score Matching), are now defined at first mention in the text.

Comment:

Correct the typo “uninfected group” to “unvaccinated group.”

Response:

This typographical error has been corrected throughout the manuscript.

Comment:

Consider adding 95% confidence interval ribbons or error bars to trend figures.

Response:

Trend figures have been updated to include 95% confidence intervals where appropriate to improve clarity and interpretability

We appreciate the reviewers’ suggestions, which have strengthened the manuscript.

---

## [Editor Report · Decision Letter 2]

22 Dec 2025

A Prospective Evaluation of Johnson & Johnson COVID-19 Vaccine on Glycemic Biomarkers in Type 2 Diabetes Mellitus in Ethiopia

PONE-D-25-35535R2

Dear Dr. Edae,

We’re pleased to inform you that your manuscript has been judged scientifically suitable for publication and will be formally accepted for publication once it meets all outstanding technical requirements.

Kind regards,

Timotius Ivan Hariyanto, M.D.

Academic Editor

PLOS One
---

## [Editor Report · Acceptance letter]

PONE-D-25-35535R2

PLOS One

Dear Dr. Edae,

I'm pleased to inform you that your manuscript has been deemed suitable for publication in PLOS One. Congratulations! Your manuscript is now being handed over to our production team.

Kind regards,

on behalf of

Dr. Timotius Ivan Hariyanto

Academic Editor

PLOS One